# Obstructive Sleep Apnea Is Associated with an Increased Risk of Developing Gastroesophageal Reflux Disease and Its Complications

**Xiaoliang Wang** [1] 🆔, **Zachary Wright** [1] 🆔, **Jiayan Wang** [1] **and Gengqing Song** [2],*

1   Internal Medicine Residency Program, Joan C. Edwards School of Medicine, Marshall University, Huntington, WV 25701, USA; wangxi@marshall.edu (X.W.); wright476@marshall.edu (Z.W.)
2   Department of Gastroenterology and Hepatology, Metrohealth Medical Center, Case Western Reserve University, Cleveland, OH 44106, USA
*   Correspondence: gsong@metrohealth.org; Tel.: +1-216-778-4123

**Abstract:** Patients with obstructive sleep apnea (OSA) commonly report gastroesophageal reflux disease (GERD) symptoms, and limited data suggest a relationship between OSA and GERD-related complications. To investigate this association, we performed a population-based analysis using National Inpatient Sample (NIS) data for 7,159,694 patients. After adjusting for risk factors, OSA patients had a significantly higher incidence of GERD (32.3%) compared to those without OSA (15.0%, $p < 0.01$). OSA patients also had a higher risk of developing GERD-related complications, including non-erosive esophagitis, erosive esophagitis, esophageal stricture, and Barrett's esophagus with and without dysplasia. Therefore, our results emphasize the importance of early detection and management of GERD and its complications in patients with OSA, particularly those with additional risk factors such as obesity and smoking.

**Keywords:** OSA; GERD; esophageal stricture; Barrett's esophagus; high risk

## 1. Introduction

Gastroesophageal reflux disease (GERD) is defined as a disorder of reflux of gastric contents into the esophagus causing symptoms such as heartburn and regurgitation. The presence of characteristic mucosal injury can also describe GERD under endoscopy or abnormal esophageal acid exposure found on a reflux monitoring testing [1,2]. Other than classic symptoms, GERD can cause extraesophageal symptoms such as hoarseness, wheezing, asthma, chest pain, and globus sensation [3]. Understanding the risk factors associated with GERD can help individuals identify their susceptibility to the disease and take preventive measures. One of the primary risk factors for GERD is obesity, as excess body weight, especially around the abdomen, increases intra-abdominal pressure and puts pressure on the lower esophageal sphincter (LES). The LES is responsible for preventing the backflow of stomach acid into the esophagus, and when the pressure on the LES is elevated, it becomes weaker and less effective, allowing acid reflux to occur more easily. Dietary factors also play a role in GERD risk as consumption of diets high in fatty foods, spicy foods, citrus fruits, chocolate, caffeine, and carbonated beverages has been associated with an increased likelihood of experiencing acid reflux. These foods can relax the LES and stimulate the production of stomach acid, exacerbating symptoms of GERD. Smoking is another significant risk factor for GERD [4]. Smoking weakens the LES and reduces saliva production, which normally helps neutralize stomach acid. Consequently, smokers are more susceptible to acid reflux and its complications. Certain lifestyle habits, such as lying down immediately after eating, snacking before bedtime, and wearing tight clothing around the waist, can also contribute to GERD risk. These behaviors can increase intra-abdominal pressure and disrupt the normal digestion process,

making acid reflux more likely. Additionally, certain medical conditions have been found to contribute to an increased GERD risk [1,5]. Hiatal hernia, a condition in which part of the stomach pushes upward through the diaphragm, can weaken the LES and facilitate acid reflux. Asthma, diabetes, and connective tissue disorders have also been linked to an increased risk of GERD. It is important to note that while these risk factors can contribute to the development of GERD, not everyone with these factors will necessarily develop the disease [1]. However, individuals who possess these risk factors should be aware of their potential susceptibility and take appropriate measures, such as adopting a healthy lifestyle, maintaining a healthy weight, and avoiding triggers to reduce the likelihood of experiencing GERD symptoms and complications [6]. The diagnosis of GERD is commonly based either on the response of suspected reflux-related symptoms to empiric acid-suppressive therapy or on an objective finding by diagnostic exams [3,7]. In the U.S., the prevalence of GERD is estimated anywhere between 18 and 28% [8], which results in a significantly impaired quality of life and a high insurance burden.

Obstructive sleep apnea (OSA) is a sleep-related breathing disorder that involves a decrease or complete block in airflow despite an ongoing effort to breathe [9]. Most patients with OSA report daytime sleepiness, or their partner reports loud snoring, choking, or breathing interruption during sleep [10,11]. These symptoms are also commonly detected while evaluating other diseases or during health maintenance screening. By stringent definitions, the estimated prevalence of OSA in the USA is approximately 15% in males and 5% in females [12,13]. The prevalence of OSA also varies by race and is more prevalent in African Americans [12]. While the impact of OSA on one's quality of sleep and overall health is well-documented, it also poses several risk factors that can have significant consequences. One of the primary risks associated with OSA is daytime sleepiness and impaired cognitive function [14]. Individuals with OSA often experience excessive daytime sleepiness, which can lead to decreased alertness, difficulty concentrating, and impaired performance at work or school. This can increase the risk of accidents and injuries in occupational and everyday settings. OSA is also closely linked to various cardiovascular conditions [15], with recurrent interruptions in breathing causing oxygen levels to drop and putting additional stress on the cardiovascular system. Consequently, OSA has been associated with high blood pressure, heart disease, heart failure, and an increased risk of stroke. Furthermore, the disrupted sleep patterns in OSA can contribute to metabolic abnormalities, such as insulin resistance and diabetes [16]. Untreated OSA can have a detrimental impact on mental health as well. Studies have shown that individuals with OSA are at a higher risk of developing depression, anxiety disorders, and even cognitive decline over time [17,18]. In addition, OSA can pose risks during surgery and anesthesia by complicating the administration of anesthesia and leading to respiratory complications post-surgery. With these risk associations in mind, it is crucial to recognize and address the risk factors associated with OSA early on. Seeking proper diagnosis and treatment can not only improve sleep quality but also reduce the risk of associated health complications, enhance daytime functioning, and ultimately promote a healthier and safer life.

Several epidemiologic studies have shown that nighttime heartburn is common and that individuals who experience heartburn at night also report sleep disturbances resulting in alteration in daytime performance [19–21]. GERD and OSA, one of the most common sleep disorders, are thought to affect each other, but the exact relationship has never been clearly demonstrated. There is some evidence showing a link between GERD and OSA. Bortolotti et al., showed that the frequency of OSA symptoms significantly decreased with omeprazole treatment in a patient with GERD [22]. In another study involving 140 patients, Friedman et al., showed that treatment of GERD with esomeprazole significantly reduced snoring and daytime sleepiness [23]. A systematic review of 20 articles by Regenbogen et al., demonstrated that treatment for GERD with proton pump inhibitors improved the quality of life in patients suffering from sleep disturbance [24].

Limited data suggest a relationship between OSA and GERD. Green et al., demonstrated that the prevalence of GERD was 62% in 331 patients with OSA. The patient

compliant with continuous positive airway pressure (CPAP) had a significant improvement in GERD symptoms as compared to no improvement in a patient not using CPAP [25]. Another study involving a small group of patients showed that using nasal CPAP significantly reduced acid reflux frequency, duration, and the percentage of time pH was less than 4 [26]. However, other studies have failed to demonstrate a significant relationship between GERD and OSA. In a study involving more than 1000 patients from sleep disorder centers, GERD was found not correlated with OSA, and the severity of OSA did not affect the prevalence of GERD [27]. Additionally, Morse et al., suggested that while patient reports sleep quality was positively affected by GERD severity, no definite correlation was found between GERD and OSA [28]. These results may be due to the fact that GERD and OSA commonly share similar risk factors, which could be confounding factors or due to the small patient sample.

This study aimed to determine whether OSA was associated with a higher prevalence of GERD by using an extensive nationwide database. We also aimed to assess the relationship between OSA and GERD complications, including esophageal stricture, Barrett's esophagus with or without dysplasia, and esophageal cancer.

## 2. Materials and Methods

### 2.1. Database

A retrospective analysis was performed using the 2017 National Inpatient Sample (NIS) database, which was developed by the Healthcare Cost and Utilization Project (HCUP). NIS is the largest publicly available all-payer inpatient healthcare database designed to estimate inpatient utilization, access, cost, quality, and outcomes, which contains unweighted data from around 7 million hospital stays each year. The NIS approximates a 20% stratified sample of all discharges from US community hospitals, excluding rehabilitation and long-term acute care hospitals.

### 2.2. Data Collection and Outcomes

A total of 7,159,694 adult patients admitted to the hospital in 2017 were included in this study. Patients diagnosed with GERD (ICD-10-CM K21.9 and K21.0) with and without OSA (ICD-10-CM G47.33) were compared to patients without GERD. We excluded subjects with a history of foregut surgeries, uncontrolled type 2 diabetes (T2DM), eosinophilic esophagitis, and infective esophagitis. OSA was based on polysomnography and excluded other etiology of sleep apnea, such as central sleep apnea, idiopathic sleep-related nonobstructive alveolar hypoventilation, or periodic high-altitude breathing. The risk factors of OSA, including male sex, being elderly, obesity, smoking history, and risk factors of GERD, including hiatal hernia, cigarette smoking, and obesity, were used for variable adjustment analysis. Based on the risk factors associated with gastroesophageal reflux disease (GERD) and obstructive sleep apnea (OSA), previous studies may not accurately reflect the true relationship between these conditions. In order to mitigate bias, we have made adjustments to the risk factors of both GERD and OSA in our study. Furthermore, we have taken into consideration the history of psychological abnormalities, such as anxiety and depression, and adjusted for psychological problems as well. We aim to provide a more comprehensive analysis of the potential link between GERD and OSA by addressing these confounding factors. However, due to the complex nature of these disorders and the various factors involved, it is essential to account for potential confounders in order to obtain a clearer understanding of their interplay. Demographic data were collected, including age, race, gender, mental health history, obesity, and smoking history. GERD complications, including esophageal stricture, Barrett's esophagus with and without dysplasia, and esophageal cancer, were included only in a patient with the diagnosis of the GERD group. To assess the odds ratio of GERD and related complications in OSA, we included those with GERD and GERD complications (cases) compared with those without the diagnosis of GERD and GERD complications (controls). All diagnoses included or excluded from this study were selected by the ICD-10-CM code.

*2.3. Statistical Analysis*

All demographic and risk factor data in this study collected from NIS were categorical and thus were presented as several cases and percentages. Chi-squared analysis was used to analyze the association between GERD and OSA and investigate the association between GERD complications in those with and without OSA. Multivariate logistic regression analysis was used to assess the risk of, in the form of odds ratios, GERD and GERD complications with and without the different stages of OSA. We adjusted for age, gender, race, cigarette smoking, and obesity as covariates to mitigate the effect of confounding factors. A 2-sample test for equal proportions was used, and a *p*-value < 0.05 was considered significant. IBM SPSS 28.0.1.1 was used for the statistical analysis.

## 3. Results

A total of 7,159,694 patients hospitalized during 2017 were included in this study, in which 1,179,759 subjects diagnosed with GERD with and without OSA were identified (138,805 with OSA and 1,040,939 without OSA) (Figure 1). Overall, GERD patients with OSA were slightly younger than those without (63.5 ± 0.04 vs. 64.4 ± 0.02, *p* < 0.05). There were significantly more males in the GERD with OSA group compared to the GERD without OSA group (49.4% vs. 40.5%, *p* < 0.01). Obesity was significantly higher in those with GERD and OSA than those without OSA (47.8% vs. 16.3%, *p* < 0.01). A higher proportion of patients with GERD and OSA also had mental health issues, such as a single episode of major depression disorder (MDDSE) (24.9% vs. 18.8% *p* < 0.01) and anxiety (24.2% vs. 21.4%, *p* < 0.01) compared to GERD without OSA. There were significantly fewer cigarette smokers in the GERD with OSA group than in the GERD without OSA group (11.3% vs. 14.8%, *p* < 0.01) (Table 1).

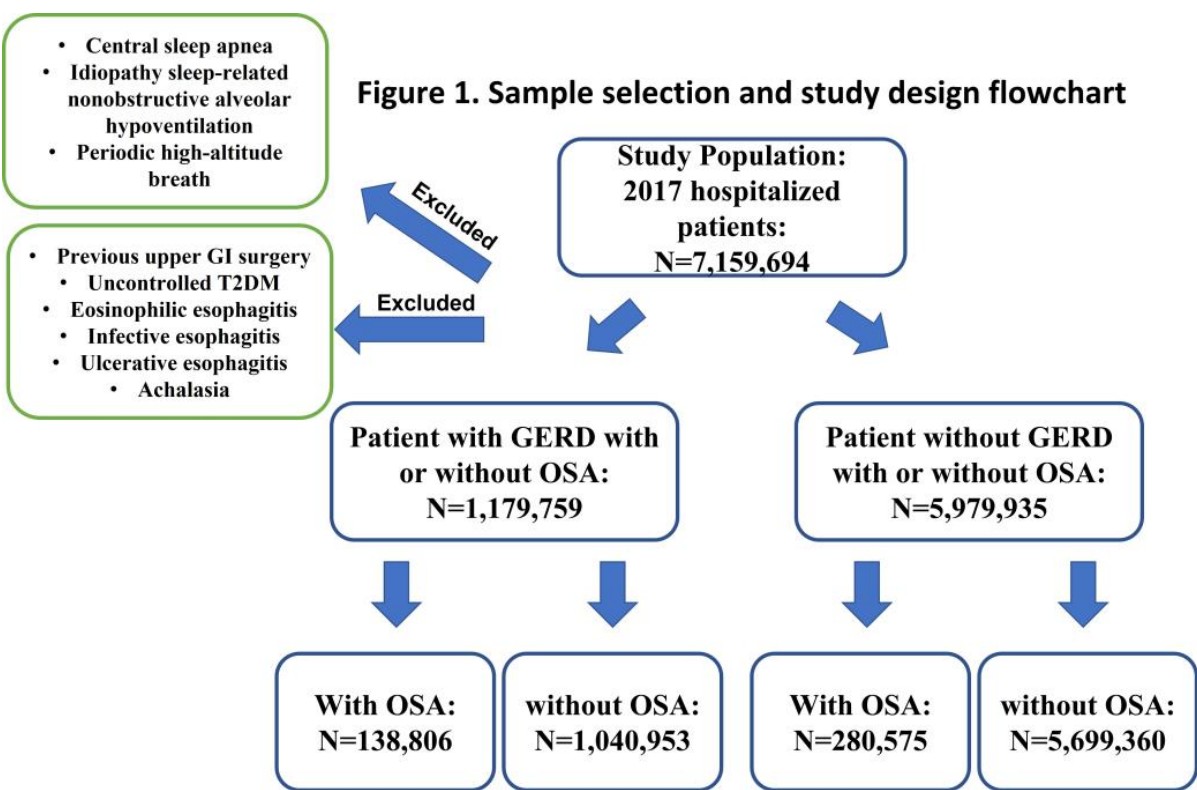

**Figure 1.** Sample selection and study design flowchart.

**Table 1.** Demographic characterization, risk factors and comorbidities between sleep apnea with and without GERD.

|  | GERD w/OSA | GERD w/o OSA | *p*-Value |
|---|---|---|---|
| Age | 63.5 ± 0.04 | 64.4 ± 0.02 | <0.05 |
| Sex |  |  | <0.01 |
| Female | 70,028 (50.5%) | 619,462 (59.5%) |  |
| Male | 68,777 (49.5%) | 421,454 (40.5%) |  |
| Race |  |  | <0.01 |
| White | 103,845 (74.8%) | 761,292 (73.1%) |  |
| Black | 18,433 (13.3%) | 126,231 (12.1%) |  |
| Hispanic | 7332 (5.3%) | 74,277 (7.1%) |  |
| Asian | 1148 (0.8%) | 17,213 (1.7%) |  |
| Comorbidity |  |  |  |
| MDD | 3524 (2.1%) | 19,724 (2.0%) | >0.05 |
| MDDSE | 41,605 (24.9%) | 183,740 (18.8%) | <0.01 |
| Anxiety | 40,383 (24.2%) | 209,429 (21.4%) | <0.01 |
| Obesity | 66,393 (47.8%) | 171,920 (16.5%) | <0.01 |
| SMOKING | 15,742 (11.3%) | 154,060 (14.8%) | <0.01 |
| Hiatal Hernia | 7765 (5.6%) | 56,958 (5.5%) | <0.05 |

GERD, gastroesophageal reflux disease; MDD, major depressive disorder; MDD, major depression disorder; MDDSE, major depression disorder single episode; w/, with; w/o, without.

Subjects with OSA were significantly more like to have non-erosive reflux disease (NERD) than those without OSA (OR 1.853, 95% CI 1.840–1.867, *p* < 0.001). The incidence of NERD in those with OSA was 3234.5 per 10,000 cases, and the incidence in those without OSA was 1497.6 per 10,000 cases (*p* < 0.001). The incidence of erosive esophagitis (EE) in OSA was 78.4 per 10,000 patients, and the incidence in those without OSA was 48.1 per 10,000. The risk of having EE was significantly high in patients with OSA (OR: 1.207, 95% CI: 1.161–1.255, *p* < 0.001). (Table 2 and Figure 2).

**Table 2.** Prevalence and odds ratios for OSA patients with GERD and GERD-related complications.

|  | | NERD | | |
|---|---|---|---|---|
| **OSA** | **Cases** | **Pevalence** | **Adjusted OR** | ***p*-Value** |
| Yes | 135,647 | 32.30% ** | 1.853 | <0.001 |
| No | 1,009,396 | 15.0% |  |  |
|  | | EE | | |
| OSA | Cases | Cases per 10,000 patients | Adjusted OR |  |
| Yes | 3287 | 78.4 ** | 1.207 | <0.001 |
| No | 32,422 | 48.1 |  |  |
|  | | GERD + ESOS | | |
| OSA | Cases | Cases per 10,000 patients | Adjusted OR |  |
| Yes | 606 | 21.4 ** | 1.565 | <0.001 |
| No | 6276 | 10.9 |  |  |

**Table 2.** *Cont.*

| NERD | | | | |
|---|---|---|---|---|
| **OSA** | **Cases** | **Pevalence** | **Adjusted OR** | **_p_-Value** |
| GERD + ESOC | | | | |
| OSA | Cases | Cases per 10,000 patients | Adjusted OR | |
| Yes | 209 | 7.4 ** | 1.350 | <0.001 |
| No | 2248 | 3.9 | | |
| GERD + BARR | | | | |
| OSA | Cases | Cases per 10,000 patients | Adjusted OR | |
| Yes | 1837 | 64.6 ** | 2.117 | <0.001 |
| No | 10,764 | 18.8 | | |
| GERD + BARR WITHOUT DYSPLASIA | | | | |
| OSA | Cases | Cases per 10,000 patients | Adjusted OR | |
| Yes | 1799 | 63.3 ** | 2.113 | <0.001 |
| No | 10,559 | 18.4 | | |
| GERD + BARR WITH DYSPLASIA | | | | |
| OSA | Cases | Cases per 10,000 patients | Adjusted OR | |
| Yes | 38 | 1.3 ** | 2.259 | <0.001 |
| No | 207 | 0.4 | | |

GERD, gastroesophageal reflux disease; NERD, non-erosive reflux disease; EE, erosive esophatitis; ESOS, esophagus stricture; ESOC, esophageal cancer, BARR, Barrett esophagus. Adjusted for age, race, gender, obesity, smoking history, hiatal hernia. ** $p < 0.01$.

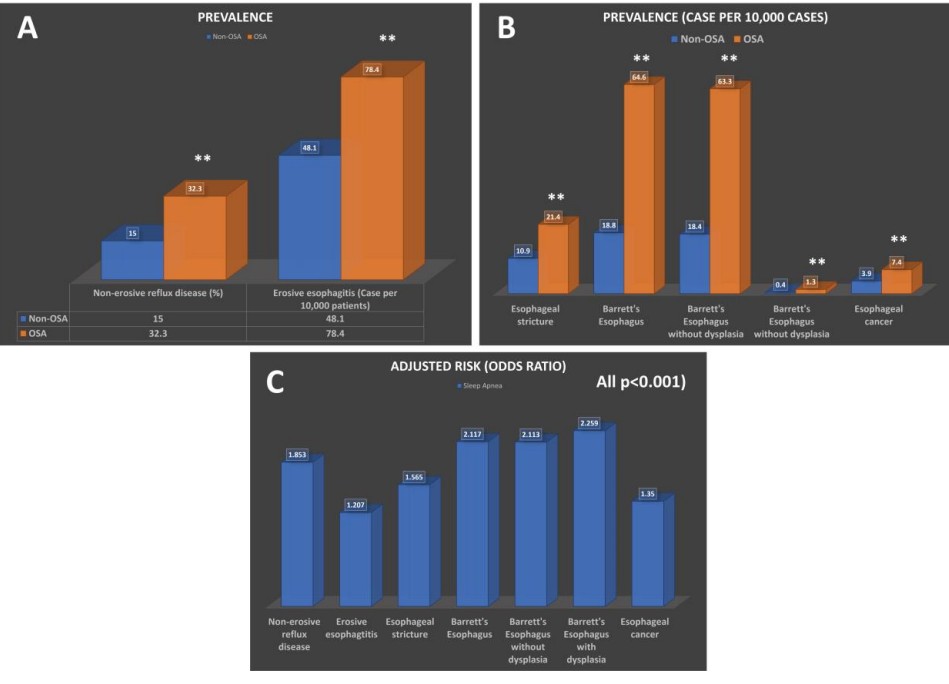

**Figure 2.** Bar graph of prevalence and odds ratio for OSA patients with GERD and GERD related complications (**A**), Prevalence of non-erosive reflux disease and erosive esophagitis in patients with OSA. (**B**), Prevalence of GERD complications including esophageal stricture, Barrett's esophagus with or without dysplasia and esophageal cancer. (**C**), Adjusted odds ratio of GERD and its complications in patients with OSA. GERD, gastroesophageal reflux disease; OSA, obstructive sleep apnea. Adjust for age, sex, race, obesity, hiatal hernia and history of smoking. ** $p < 0.01$.

Among complications potentially associated with GERD, the risk of esophageal stricture was significantly higher in subjects with OSA (OR 1.565, 95% CI 1.431–1.711, $p < 0.001$). The incidence of esophageal stricture in OSA was 21.4 per 10,000 patients, and the incidence in those without OSA was 10.9 per 10,000 ($p < 0.01$) (Table 2 and Figure 2).

Subjects with OSA and GERD were significantly more likely to develop Barrett's esophagus compared to those without OSA (OR 2.117, 95% CI: 2.005–2.235, $p < 0.001$), with an incidence of 64.6 per 10,000 subjects among those with OSA and the incidence of 18.8 per 10,000 subjects among patient without OSA ($p < 0.01$). Sub-analysis showed that the incidence of Barrett's esophagus without dysplasia was 63.3 per 10,000 patients among those with OSA and 18.4 per 10,000 patients among patients without OSA ($p < 0.01$). The risk of Barrett's esophagus without dysplasia was much higher in patients with OSA (OR: 2.113, 95%CI: 2.000–2.232, $p < 0.01$). We also found that in patients with dysplasia, the incidence was 1.3 per 10,000 patients in OSA subjects compared to 0.4 per 10,000 patients in those without OSA ($p > 0.01$). The risk of Barrett's esophagus with dysplasia was significantly higher in subjects with OSA (OR: 2.259, 95%CI: 1.557–3.277, $p < 0.01$) (Table 2 and Figure 2).

Furthermore, the risk of esophageal cancer was significantly higher in subjects with OSA (OR 1.350 95% CI 1.161–1.569, $p < 0.01$). The incidence of esophageal cancer in OSA was 7.4 per 10,000 patients, and the incidence in those without OSA was 3.9 per 10,000 ($p < 0.01$) (Table 2 and Figure 2).

## 4. Discussion

The most important findings in this study are the significantly increased risk and incidence of GERD and its complications in those with OSA compared to those without OSA. This is the first study that specifically used extensive inpatient patient data to investigate the association between GERD and OSA and demonstrated a clear association. Our finding is even more significant as we have adjusted several important possible confounding factors, such as obesity, smoking, male sex, mental health problems, and age.

Mahfouz et al., using the same database, have shown a higher incidence of GERD in OSA patients than in patients without GERD. Specifically, they found that in patients diagnosed with GERD, 12.21% also had a concurrent diagnosis of OSA, which was significantly higher compared to the 4.79% of patients without GERD. The mean age of patients with GERD and OSA was slightly lower than those without OSA. Regarding comorbidities, patients with GERD and OSA had a higher prevalence of obesity, diabetes mellitus, hypertension, atrial fibrillation, congestive heart failure, and pulmonary hypertension [29]. Still, an association study was not conducted between GERD and OSA [30]. Other studies have shown the improvement of GERD symptoms in OSA patients using CPAP and found a strong correlation between CPAP pressure and improvement of GERD symptoms [25,26]. In another long-term prospective follow-up study investigating the relationship between OSA and nocturnal gastroesophageal reflux (nGER) as well as the impact of continuous positive airway pressure (CPAP) treatment on nGER symptoms. A cohort of 331 OSA patients diagnosed from 1993 to 2000, was prospectively examined. Baseline assessments included patients self-reporting the frequency of nGER symptoms. All patients were prescribed CPAP therapy for their OSA. Prior to CPAP treatment, 62% of the OSA patients experienced nGER. Out of the 181 patients who provided follow-up data, 91% were adherent to CPAP treatment, while 9% were non-compliant, forming the treatment and control groups, respectively. Patients who consistently used CPAP experienced a significant improvement in their nGER scores (48% improvement), with a mean reduction from 3.38 before CPAP treatment to 1.75 after. Conversely, the control group (non-CPAP users) showed no significant improvement in nGER scores. The study also found a strong correlation between CPAP pressure levels and improvement in nGER scores (correlation, r = 0.70; $p < 0.001$), indicating that higher CPAP pressures were associated with greater improvement in nGER symptoms [25].

However, a recent study conducted at the University of Arizona Sleep Center involved 136 patients referred for polysomnographic studies. The patients were evaluated using a demographic survey, the GERD Symptom Checklist, and the Sleep Heart Health Study Sleep Habits Questionnaire. Polysomnograms were used to assess objective measures of sleep and breathing. They found that the severity of sleep apnea was not related to self-reported heartburn or acid regurgitation symptoms or the severity of GERD. Objective measures of disordered sleep were more strongly associated with age, smoking, and alcohol use in men and with age and body mass index in women compared to GERD. GERD severity had a greater impact on subjectively reported sleep quality than age, smoking, alcohol use, or the presence of OSA. However, only females taking anti-reflux medications were less likely to report poor sleep quality [31]. Additionally, a study involving 17 patients with sleep apnea divided them into two groups based on sleep apnea severity. Both groups showed a high occurrence of GERD but no significant difference in reflux times between distal and proximal esophageal sites. Reflux episodes and apnea periods were not correlated. Most patients in both groups were obese. This study suggests a link between sleep apnea and GERD, regardless of sleep apnea severity, and highlights obesity as a common risk factor. A similar finding was reported by Morse et al., who found no objective correlation between OSA and GERD in a study involving 136 subjects [28].

Importantly, we have also shown that several GERD-associated complications affect OSA patients more frequently than those without OSA. These include esophageal stricture, Barrett's esophagus, and esophageal cancer. We found that OSA patients had a 50% higher risk of getting esophageal stricture and a two times higher risk for Barrett's esophagus. Moreover, we found that patients with OSA had a slightly higher risk of experiencing Barrett's esophagus with dysplasia than Barrett's esophagus without dysplasia (OR: 2.113 vs. 2.259).

Along with the finding that OSA patients also had a 35% higher risk for esophageal cancer when compared to those without OSA, this may indicate that OSA could be an independent risk factor for esophageal cancer, which requires more attention [32]. While we could not demonstrate the exact pathophysiology mechanism between OSA and these complications, there was a clear association between them. A meta-analysis by Elfanagely et al., involving 2333 subjects, found a significantly increased risk of OSA among patients with Barrett's esophagus [33]. A similar finding was reported by Cummings et al., in a study that included 287 patients, where they found a high incidence of patients with Barrett's esophagus at an increased risk for OSA, and night acid flux symptoms were associated with OSA [34]. More importantly, a group at West Virginia University reported that patients with OSA had a significantly increased risk of Barrett's esophagus than patients without OSA, and the risk has positively associated with the severity of OSA [35].

The exact mechanism for why those with OSA have more chance of developing GERD and its complications is still unclear. It has been suggested that OSA might induce gastric distension, decrease gastric emptying, and cause transient reflex activation of the lower esophageal sphincter (LES) smooth muscle or transmission of pressure to the LES [36]. A study aimed to investigate the effect of continuous positive airway pressure (CPAP) on swallow-induced relaxation of the lower esophageal sphincter (LES) involved 10 healthy individuals who were awake and in a supine position. Measurements of various pressures, including esophageal pressure, LES pressure, gastric pressure, and the barrier pressure to reflux, were taken. They found that CPAP significantly increased end-expiratory esophageal pressure, LES pressure, and gastric pressure during quiet breathing. During swallowing, CPAP reduced the duration of LES relaxation. These findings suggest that CPAP may enhance LES function, making it less susceptible to reflux by increasing Pb and reducing the duration of LES relaxation. This could explain why OSA patients, who commonly experience GERD during periods of sleep apnea, experience a reduction in GERD symptoms with CPAP usage [21,36].

Moreover, OSA may increase trans-diaphragmatic pressure and reduce intra-thoracic pressure, favoring acid reflux [37]. In a study involving six obese patients with obstruc-

tive sleep apnea syndrome (OSAS), the impact of apnea on diaphragmatic function was examined, and 119 randomly selected apneas were analyzed during non-REM sleep. The results indicated a progressive increase in diaphragmatic electromyogram power spectrum, and maximum relaxation rate of transdiaphragmatic pressure during the obstructive phase within each apnea. This increased transdiaphragmatic pressure may contribute to the development of acid reflux [38]. Additionally, more significant respiratory effects after the apnea episode increase the pressure gradient across the LES and may eventually cause the retrograde movement of gastric contents [20,39]. These transient LES-relaxation-induced GERD may be mild and would not cause erosive esophagitis. This is consistent with our finding that patients with OSA had an 80% higher risk of experiencing GERD without esophagitis but only a 20% higher risk of getting GERD with esophagitis than those without OSA.

Several limitations have been identified in our study. NIS, a nationwide inpatient database, collected all data, but outpatient information, which accounts for another part of clinical practice, was not included. The diagnosis of GERD and each complication assessed in this study was based on ICD-10-CM codes, which were entered among various hospital systems and electronic medical records. It is assumed that the diagnosis of each complication of GERD, including esophageal stricture, Barratt's esophagus, and esophageal cancer, was based on images, endoscopy, and pathology. The risk factors of GERD, such as smoking, diabetes, and hiatal hernia, were determined via ICD-10-CM codes as well; no timeline for these risk factors could be identified. Lastly, we also used ICD-10-CM codes to select subjects with different stages of OSA, which is assumed to be diagnosed based on polysomnography. The present retrospective database-dependent study has certain limitations that should be acknowledged. Firstly, one important limitation is the absence of medication history in the database. Medications can potentially contribute to the development of GERD induced by OSA [40]. Therefore, more information regarding medication usage is needed to ensure a comprehensive analysis of potential alternative etiologies of OSA-induced GERD. Future studies should consider incorporating medication history to understand better the relationship between OSA, medication use, and GERD. Secondly, the database used in this study does not provide disease timeline information. This absence of data restricts the ability to conduct a correlation study between OSA and GERD. Disease timelines would allow for a more precise determination of whether GERD preceded or followed the diagnosis of OSA [41]. This temporal information is crucial in establishing a potential causal relationship between these conditions. Therefore, without disease timeline data, the study is unable to establish a cause-and-effect relationship between OSA and GERD. Future investigations should further endeavor to include disease timeline information to elucidate the temporal relationship between these two conditions. Recognizing the current study's limitations, it is essential to address these gaps through additional research. To mitigate the aforementioned limitations, a prospective study is being planned that will examine data from regional hospitals over the past 7 years. This future study aims to confirm the current findings and address some of the partial limitations of the retrospective study. By including medication history and disease timeline information in the new investigation, a more comprehensive analysis of the relationship between OSA and GERD can be conducted. The prospective study will provide valuable insights into the potential etiology and temporal.

The clinical implications of our study are that patients with OSA should be investigated for the presence of GERD and GERD-related complications, especially for Barrett's esophagus with dysplasia and esophageal cancer. Early treatment of GERD with antacids or screening upper endoscopy in patients with OSA should be considered to prevent GERD complications.

**Author Contributions:** X.W.: conceptualization, investigation, data curation, data analysis, visualization, drafting the manuscript. Z.W.: editing manuscript. J.W.: editing manuscript. G.S.: conceptualization, project supervision, editing manuscript. All authors have read and agreed to the published version of the manuscript.

**Funding:** This research received no external funding.

**Institutional Review Board Statement:** The Marshall University School of Medicine Institutional Review Board has deemed studies using the NIS database as exempt from requiring IRB approval due to the de-identified and aggregated nature of the data in the database. The Case Western Reserve University/Metrohealth Medical Center Institutional Review Board has deemed studies using the NIS database as exempt from requiring IRB approval due to the de-identified and aggregated nature of the data in the database at the standard defined in Section 164.514(a) of the HIPAA Privacy Rule.

**Informed Consent Statement:** Patient consent was waived due to no patient's identity being collected in this database.

**Data Availability Statement:** The data presented in this study are available on request from the corresponding author. The data are not publicly available due to patient and hospital information privacy and the requirement of H.CUP.

**Conflicts of Interest:** The authors declare no conflict of interest.

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
