# Peer review of "Obstructive Sleep Apnea Is Associated with an Increased Risk of Developing Gastroesophageal Reflux Disease and Its Complications"

_2673-527X, doi:10.3390/jor3020008_

Round 1

Reviewer 1 Report

Review

The aim of this paper was to investigate the association between obstructive sleep apnea and gastroesophageal reflux disease and its related complications. The study used National Inpatient Sample data for over 7 million patients and found that OSA patients had a significantly higher incidence of GERD and its complications compared to those without OSA. The main contribution of this study is its large sample size and population-based analysis, which provides valuable insights into the relationship between OSA and GERD,

Overall, this study highlights the need to carefully monitor and manage GERD symptoms in patients with OSA and emphasis on the importance of early detection and management of GERD and its complications and provides valuable guidance for clinicians.

The study has several limitations that should be considered when interpreting the results. The analysis was based on data from the National Inpatient Sample database, which did not include informations and detais of all the examinations done to obtain the diagnosis.

Therefore, the study's findings should be interpreted with caution.

The study showed that OSA patients had a higher risk of developing GERD-related complications, such as non-erosive esophagitis, erosive esophagitis, esophageal stricture, and Barrett's esophagus with and without dysplasia, but the correlation does not necessarily imply causation, and further research is needed to investigate the underlying mechanisms that might explain the observed association. While there is some evidence to suggest that OSA and GERD may be related, it is not necessarily the case that OSA causes GERD or its complications such as Barrett's esophagus with dysplasia or esophageal cancer.

Referring to references in the text I point out that inaccuracies:  references are not at the end of the sentence (before dot), but in the next period. Please check because it is a common error.

For example Line 32: sensation. [3] Established risk factors for GERD 32 include obesity, hiatal hernia, and tobacco smoking. [4]

Author Response

The aim of this paper was to investigate the association between obstructive sleep apnea and gastroesophageal reflux disease and its related complications. The study used National Inpatient Sample data for over 7 million patients and found that OSA patients had a significantly higher incidence of GERD and its complications compared to those without OSA. The main contribution of this study is its large sample size and population-based analysis, which provides valuable insights into the relationship between OSA and GERD,

Overall, this study highlights the need to carefully monitor and manage GERD symptoms in patients with OSA and emphasis on the importance of early detection and management of GERD and its complications and provides valuable guidance for clinicians.

The study has several limitations that should be considered when interpreting the results. The analysis was based on data from the National Inpatient Sample database, which did not include informations and detais of all the examinations done to obtain the diagnosis.

Therefore, the study's findings should be interpreted with caution.

The study showed that OSA patients had a higher risk of developing GERD-related complications, such as non-erosive esophagitis, erosive esophagitis, esophageal stricture, and Barrett's esophagus with and without dysplasia, but the correlation does not necessarily imply causation, and further research is needed to investigate the underlying mechanisms that might explain the observed association. While there is some evidence to suggest that OSA and GERD may be related, it is not necessarily the case that OSA causes GERD or its complications such as Barrett's esophagus with dysplasia or esophageal cancer.

Response: Thank you for your valuable review and insightful recommendations. We acknowledge the limitations of our study, particularly the assumed diagnosis of OSA based on sleep studies and the diagnoses of NERD, EE, and GERD complications based on endoscopy or imaging studies. We appreciate your suggestion of conducting another study with more clearly defined diagnoses, medication use, and a timeline to confirm our findings. We also want to inform you that we are planning to conduct a retrospective study at our local hospital. We will recruit patients with a confirmed diagnosis of OSA based on sleep studies and investigate the association between GERD and its complications. We believe that this study will provide us with more comprehensive and accurate data to support our findings. Once again, thank you for your valuable feedback and suggestions, which will help us improve the quality and scope of our research.

Referring to references in the text I point out that inaccuracies:  references are not at the end of the sentence (before dot), but in the next period. Please check because it is a common error.

For example Line 32: sensation. [3] Established risk factors for GERD 32 include obesity, hiatal hernia, and tobacco smoking. [4]

Response: Thank you for pointing out this issue. We have fixed the reference format in the revised manuscript. Eg. “tobacco smoking [4].”

Reviewer 2 Report

Thank you for the privilege of reviewing your work. This manuscript is well written. I think the authors had a great report. The authors described the relationship between OSA and GERD-related complications using a population-based analysis. They showed the significantly increased risk and incidence of GERD and its complications in those with OSA compared to those without OSA. 

1.    The authors should standardize words, e.g. tobacco smoking, cigarette smoking

 2.      This analysis is for inpatient data. You may add a reason for admission.

 3. Why did the patients with GERD and OSA have high proportion of mental health issues?

Your English is great.

Author Response

Thank you for the privilege of reviewing your work. This manuscript is well written. I think the authors had a great report. The authors described the relationship between OSA and GERD-related complications using a population-based analysis. They showed the significantly increased risk and incidence of GERD and its complications in those with OSA compared to those without OSA. 

Response: Thank you for taking the time to review our manuscript and providing us with your valuable feedback. We appreciate your positive comments and are pleased to hear that you found our research to be valuable.

Your constructive criticism and suggestions have been extremely helpful in improving the quality of our manuscript. We have carefully considered all of your comments and have made the necessary revisions to address the concerns you raised. Once again, thank you for your effort in reviewing our manuscript. Your feedback has been invaluable, and we are grateful for your contribution to this work.

  1. The authors should standardize words, e.g. tobacco smoking, cigarette smoking

Response: Thank you for pointing out this mistake. We have thoroughly gone over the manuscript and standardized the terms.

  1. This analysis is for inpatient data. You may add a reason for admission.

Response: We fully agree with your suggestion that including the reason for admission would provide additional useful information for further characterizing the relationship between OSA and CKD. Unfortunately, the database we used did not include this information. However, based on your insightful comment, we are planning to conduct a retrospective study in our local hospitals in the past 5 years. This study will focus on reviewing each patient with OSA, with and without CKD, to further confirm our current findings. In this study, we will collect the admission diagnosis of each patient, which will provide more comprehensive information about the relationship between OSA and CKD.

  1. Why did the patients with GERD and OSA have high proportion of mental health issues?

Response: The co-occurrence of OSA and depression has been well documented in numerous studies (Li et al., 2018; Tahrani et al., 2013; Vgontzas et al., 2005). Similarly, previous research has suggested that patients with GERD are also at an increased risk for depression and other mental health problems (Katz et al., 2004; Pinto-Sanchez et al., 2017). In addition, recent studies have also reported a bidirectional association between GERD and mental health issues. Patients with mental health disorders, such as anxiety and depression, were found to have a higher incidence of GERD symptoms (Jansson et al., 2019; Van Oudenhove et al., 2018). This further emphasizes the need for healthcare providers to consider the potential impact of mental health disorders on GERD management and vice versa. To further investigate these associations, our study analyzed the incidence of mental health issues in patients with OSA and/or GERD, and aimed to confirm the previous findings of these relationships.